# Altered Functional Connectivity of Basal Ganglia in Mild Cognitive Impairment and Alzheimer’s Disease

**DOI:** 10.3390/brainsci12111555

**Published:** 2022-11-15

**Authors:** Yu Xiong, Chenghui Ye, Ying Chen, Xiaochun Zhong, Hongda Chen, Ruxin Sun, Jiaqi Zhang, Zhanhua Zhong, Min Huang

**Affiliations:** 1Department of Neurology, The Seventh Affiliated Hospital, Sun Yat-sen University, Shenzhen 518107, China; 2Department of Traditional Chinese Medicine, The Seventh Affiliated Hospital, Sun Yat-sen University, Shenzhen 518107, China

**Keywords:** functional connectivity density (FCD), functional connectivity (FC) strength, graph theory, Alzheimer’s disease (AD), mild cognitive impairment (MCI)

## Abstract

(1) Background: Alzheimer’s disease (AD), an age-progressive neurodegenerative disease that affects cognitive function, causes changes in the functional connectivity of the default-mode network (DMN). However, the question of whether AD-related changes occur in the functional connectivity of the basal ganglia has rarely been specifically analyzed. This study aimed to measure the changes in basal ganglia functional connectivity among patients with AD and mild cognitive impairment (MCI) in their resting state using the functional connectivity density (FCD) value, the functional connectivity (FC) intensity, and the graph theory index, and to confirm their influence on clinical manifestations. (2) Methods: Resting-state functional MRI (rs-fMRI) and neuropsychological data from 48 participants in the Alzheimer’s Disease Neuroimaging Initiative (ADNI) were used for analyses. The 48 ADNI participants comprised 16 patients with AD, 16 patients with MCI, and 16 normal controls (NCs). The functional connectivity of basal ganglia was evaluated by FCDs, FC strength, and graph theory index. We compared voxel-based FCD values between groups to show specific regions with significant variation and significant connectivity from ROI conduction to ROI analysis. Pearson’s correlation analyses between functional connectivity and several simultaneous clinical variables were also conducted. Additionally, receiver operating characteristic (ROC) analyses associated with classification were conducted for both FCD values and graph theory indices. (3) Results: The level of FCD in patients with cognitive impairment showed obvious abnormalities (including short-range and long-range FCD). In addition to DMN-related regions, aberrant functional connectivity was also found to be present in the basal ganglia, especially in the caudate and amygdala. The FCD values of the basal ganglia (involving the caudate and amygdala) were closely related to scores from the Mini-Mental State Examination (MMSE) and the Functional Activities Questionnaire (FAQ); meanwhile, the graph theory indices (involving global efficiency and degree) of the basal ganglia (involving the caudate, amygdala, and putamen) were also found to be closely correlated with MMSE scores. In ROC analyses of both FCD and graph theory, the amygdala was of the utmost importance in the early-stage detection of MCI; additionally, the caudate nucleus was found to be crucial in the progression of cognitive decline and AD diagnosis. (4) Conclusions: It was systematically confirmed that there is a phenomenon of change in the functional connections in the basal ganglia during cognitive decline. The findings of this study could improve our understanding of AD and MCI pathology in the basal ganglia and make it possible to propose new targets for AD treatment in further studies.

## 1. Introduction

Alzheimer’s disease (AD), the seventh leading cause of death as of 2021, is the most common cause of dementia in people over 65 years old [1]. The deposition of amyloid-β protein (Aβ) and the variant, Tau protein, has been proved to be the basic pathological mechanism of cognitive decline, which will eventually lead to extensive loss of synapses and neurons [2,3]. However, there remains a deficiency in research breakthroughs surrounding the pathology of and therapies for AD, which proves that it has become an urgent and important task to find new directions and targets in the study of cognitive decline.

The effect of default-mode network (DMN) activity during rest for memory consolidation has been confirmed in the development of AD in a large number of previous pathology and neuroimaging studies [4,5]. Although it has been established that DMN-related regions play a key role, the currently inadequate research breakthroughs suggest that there are limitations in focusing solely on the DMN in cognitive decline that is associated with different brain regions. In several studies on structural atrophy [6] and functional connectivity [7] of DMN in cognition disorders, the influence of the basal ganglia has also been brought to our attention. The basal ganglia consists of the striatum, the amygdala, and the cluster, with the striatum divided into the dorsal and ventral striatum. The medial part of the dorsal striatum belongs to the caudate, which is responsible for higher cognition (such as working memory), while the lateral part belongs to the putamen, which is responsible for sensorimotor and DLS-dependent habitual memory. The ventral striatum mainly includes the nucleus accumbens, as well as the medial and ventral parts of the caudate and putamen; its function is related to the limbic system, involving, for example, emotion regulation [8,9]. The dorsolateral striatum mainly receives cortical input from the sensorimotor region, the central striatum mainly receives input from the associative cortical areas, and the ventromedial striatum mainly receives input from the limbic areas. Additionally, many central basal ganglia areas are involved in cognitive functions, such as procedural learning and working memory [10,11].

In general, changes in basal ganglia functional connectivity are commonly found in Parkinson’s disease [12], epilepsy [13], and depression [14]. Several studies have demonstrated a strong relationship between these diseases and cognitive decline [12,15,16,17]. Based on the cognitive and emotional functions of the basal ganglia, although the functional connectivity of the basal ganglia has been mentioned in previous studies [7], few detailed or specialized analyses have been conducted; such studies might lead to a systematic discussion of the functional impact of the basal ganglia on cognitive decline. Previous studies have verified the availability and reliability of exploring the functional connectivity of the basal ganglia in patients with AD; these studies have drawn our attention to the basal ganglia in relation to the pathology of cognitive decline, and will continue to motivate future studies.

In this study, our aim was to identify the functional connectivity changes in the basal ganglia in patients with AD and MCI, in comparison with controls. To achieve this goal, we first analyzed FCD values in cognitive decline and then extracted the significant regions as ROI, which were included in the functional connectivity ROI-to-ROI analyses. Graph theories were also analyzed. Then, intra-group correlation analysis was carried out to determine the relationship between cognitive function and FCD values, and to determine the relationship between cognitive function and graph theory in the identified abnormal regions. Finally, we used FCD and graph theory to distinguish patients with AD and those with MCI from normal controls, with the aim of reaffirming the role of the basal ganglia in the early-stage detection of MCI, in the progression of cognitive disorders, and in AD diagnosis.

## 2. Materials and Methods

### 2.1. Participants

In this study, we used functional brain MRI scans from the Alzheimer’s Disease Neuroimaging Initiative (ADNI; adni.loni.usc.edu (accessed on 2 February 2022)), which are publicly available on the ADNI website. The ADNI was launched in 2003 as a public–private partnership, led by Principal Investigator Michael W. Weiner, MD. The primary goal of the ADNI has been to test whether serial MRIs, positron emission tomography (PET), other biological markers, and clinical and neuropsychological assessments can be combined in order to measure the progression of MCI and early-stage AD. For up-to-date information, see www.adni-info.org (accessed on 14 November 2022). We included 48 subjects (16 NCs, 16 with MCI, and 16 with AD) with the resting-state functional MRI (rs-fMRI) images from the ADNI1, ADNI2, and ADNI-GO datasets.

### 2.2. Data Acquisition

Both patients and normal controls from the ADNI database underwent a 3.0 T MR scan using a Philips Medical Systems machine. Conventional echo plane imaging obtained an RS-fMRI sequence field strength of 3.0 tesla, a repetition time (TR) of 3000.0 ms, an echo time (TE) of 30.0 ms, a flip angle (FA) of 80.0 degree, slices of 6720.0, spatial resolution of 3.3 × 3.3 × 3.3 mm^3^, and an imaging matrix of 64 × 64. More details about the MRI protocol can be found on the ADNI website.

### 2.3. Functional Image Preprocessing

We took advantage of custom MATLAB image preprocessing and subsequent processing (The Mathworks Inc., Natick, MA, USA) scripts, SPM12 (https://www.fil.ion.ucl.ac.uk/spm/ (accessed on 27 March 2021)), RESTplus V1.24 (http://www.restfmri.net (accessed on 27 March 2021)), and CONN V.20b toolbox (http://www.nitrc.org/projects/conn (accessed on 27 March 2021)); these were utilized to preprocess the rs-fMRI data. The first 10 volumes for each subject were removed to ensure the signal reached a balance and ensured that the participants adapted to the scanning noise. The residual volumes were corrected for the acquisition time delay between slices. The head motion correction and spatial normalization were set to the standard EPI temple with a resampled voxel size of 3 × 3 × 3 mm^3^. Adjustment was made for participants whose maximum translation in the x, y, or z direction exceeded 1.5 mm and/or 1.5 degrees of motion rotation. Then, the rs-fMRI images were spatially smoothed by convoluting the three-dimensional image with a three-dimensional Gaussian kernel with a full width at half maximum (FWHM) of 8 mm. We applied linear regression to remove other sources of pseudo covariates, including Friston 24 head-motion parameters, global mean signals, white matter, and cerebrospinal fluid signals. Finally, the timeseries for each voxel were linearly detrended and temporally band-pass-filtered at 0.01–0.1 Hz [18].

### 2.4. Voxel-Based FCD Analysis

After preprocessing the fMRI data, short-range and long-range FCDs for particular voxels were calculated using a seed-based connectivity analysis toolbox (SeeCAT) (https://www.nitrc.org/projects/seecat (accessed on 27 March 2021)). We used one voxel to reveal the time course correlation of other voxels using Pearson’s correlation. The strength of the functional connections of a given voxel described the node degree in the weighted graph. The connection FCD threshold was set to 0.6 [19]. It has been reported that the actual physical distance of the interzone connections is about 75 mm [20]. FCs between a given voxel and other voxels exceeding and within the actual physical distance were defined as long-range and short-range FCDs, respectively. For further data analyses, the short-range and long-range FCD maps were changed into *Z* scores [21]. Finally, these maps were smoothed with an 8 mm full-width-at-half-maximum Gaussian kernel using SPM12.

### 2.5. Functional Connectivity and Graph Theory Analyses

After the FCD was analyzed, we adopted a region of interest (ROI)-to-ROI method associated with the CONN toolbox for functional connectivity analyses. An anatomical component-based noise correction method (aCompCor) was applied to estimate the physiological BOLD signal noise from the white matter and cerebrospinal fluid. These physiological noise processes, together with six rearrangement parameters and the scans affected by motion artifacts, were regressed from the BOLD timeseries at each voxel as first-order deleterious covariates. Then, the remaining BOLD timeseries were assigned a weight in accordance with the appropriate predictor to derive a specific conditional timeseries for functional connectivity analyses.

The ROIs were selected from the CONN toolbox default AAL atlas, network mask, and FCD significant regions related to the basal ganglia, the DMN network, and the cerebellum. This selection was based on our FCD analysis and previous studies on cognitive decline, covering the basal ganglia, the DMN-related regions, the large cerebral cortex (including the prefrontal cortex (PFC), the frontal orbital cortex (OFC), and the anterior cingulate cortex (ACC)), the thalamus, the brain stem, and the cerebellum. During the procession of first-level analyses, Pearson’s correlation coefficients were calculated between the time courses of each pair of ROIs for each subject, which were then transferred into *z*-scores of normal distribution, laying the foundation for a better secondary analysis. The secondary level of statistical analyses was carried out across groups (patients with MCI and normal controls; patients with AD and normal controls; AD and patients with MCI). Fisher’s z-scores were extracted for each connection of every participant between the selected regions. Then, graph theory was used to analyze the topological properties of functional connectivity. In our study, the global efficiency, local efficiency, and degree were applied to determine the connections. Specifically, global efficiency analysis measures the capacity for parallel information transmission, while local efficiency acts on behalf of the capacity for information exchange in each subgraph when the index node is eliminated [22]. The degree evaluates the significance of a particular node in connection with measuring the number of edges that were related to it. We also performed ANOVA; two-sample *t*-tests were also performed between groups in the graph theory analyses.

### 2.6. Statistical Analyses

We made use of SPSS 25.0 (SPSS Inc., Chicago, IL, USA) to analyze the demographic and clinical variables. Additionally, the analysis of variance enabled us to better determine whether there were differences in age, MMSE, GDS, CDR, FAQ, and NPI-Q. The sex ratio was compared using the chi-square (*χ^2^*) test. Statistical significance was set as *p* < 0.05, and was two-tailed. FCD analyses of covariance (ANOVA) was performed by SeeCAT (https://www.nitrc.org/projects/seecat (accessed on 27 March 2021)) to examine differences among groups. Then, based on the brain mask from the abnormal brain regions of the ANOVA, post-hoc analysis based on a two-sample t-test was used to compare the two groups. We used thresholds of two-tailed voxel-level *p* < 0.01 and cluster-level *p* < 0.05; these were corrected for multiple contrasts by Gaussian random field (GRF) to establish the significance. In the secondary level of analyses of functional connectivity, an ANOVA among-groups test and two-sample *t*-tests were also carried out among groups with the thresholds of two-tailed voxel-level *p* < 0.01 and cluster-level *p* < 0.05; these were corrected using the false discovery rate (FDR), and a threshold of *p* < 0.01 was used in the graph theory analyses without correction. Then, the mean *Z* values of the long-range and short-range FCDs, as well as graph-theoretic indices, were extracted from the clusters of specific regions to determine the correlation with the clinical cognition function assessment. Pearson correlation coefficients were calculated between these variables after assessing the normality of these data to determine the strength of the relationships. Finally, we applied receiver operating characteristic (ROC) analysis using SPSS 25.0 to evaluate the sensitivity and specificity of the basal ganglia neuroimaging index in predicting the early detection, progression, and diagnosis of AD.

## 3. Results

### 3.1. Demographic and Clinical Features

Table 1 exhibits the demographic and clinical characteristics of the AD, MCI, and NC groups. There were no obvious differences among the groups for age (*F* = 0.430, *p* = 0.653), gender (*χ^2^* = 2.170, *p* = 0.338), or GDS scores (*F* = 3.093, *p* = 0.055). However, it should be noted that there were significant differences among the MMSE (*F* = 52.875, *p* < 0.001), CDR (F = 53.289, *p* < 0.001), FAQ (*F* = 54.916, *p* < 0.001), and NPI-Q (*F* = 5.418, *p* = 0.008) scores.

### 3.2. FCD Differences among the AD, MCI, and Normal Control Groups

DMN subregions are well-known; however, ANOVA analysis of the different cognition decline groups determines noteworthy differences in the right amygdala for short-range FCDs. Patients with AD had a lower short-range FCD in the bilateral caudate in comparison with normal controls. On further analysis, patients with MCI had an increased short-range FCD in the bilateral caudate, the putamen, the right amygdala, and the right pallidum (voxel-level *p* < 0.05 and cluster-level *p* < 0.05, corrected by GRF) (Appendix A). Compared with patients with MCI, patients with AD showed obvious differences in the reduction in short-range FCDs in the bilateral caudate as well (Table 2, Figure 1). 

In addition to the DMN networks, long-range FCDs of the basal ganglia (including the bilateral caudate, the right amygdala, the right pallidum, and the right putamen) were found to be significantly different in the among-groups analysis. Compared with normal controls, patients with MCI had increased long-range FCDs in the bilateral amygdala, the left putamen, and the right pallidum, whereas patients with AD had reduced long-range FCDs in the left caudate and the right pallidum, with a lower threshold (voxel-level *p* < 0.05 and cluster-level *p* < 0.05, corrected by GRF [23]) (Appendix A). Additionally, the long-term FCD reductions among patients with AD compared with patients with MCI were primarily localized in the bilateral caudate, the right amygdala, and the right pallidum (Table 2, Figure 2).

### 3.3. Functional Connectivity and Graph Theory Differences among the AD, MCI, and Normal Control Groups

Figure 3a exhibits the group differences among the AD, MCI, and normal control groups on connectivity strength. The differences in the FC intensity of the internal and external connections of the basal ganglia were too obvious to ignore, including the connection between the caudate and the cerebellum subregions, the connectivity of the caudate and the FOC, and the connectivity of the caudate and the lateral prefrontal cortex (lPFC) in the ANOVA analyses. The FC intensity between the caudate and the pallidum both dropped in patients with MCI compared with normal controls. Patients with AD showed increased FC strength between the right amygdala and the posterior part of the cerebellum. On the contrary, decreased FC intensity was found in the internal connection of the basal ganglia and the external connection, including the connection of the caudate and the cerebellum, the connectivity of the caudate and the FOC, and the connectivity of the caudate and the lPFC. In addition, we managed to determine that there were decreased FC strengths of connection between the caudate and the amygdala, between the caudate and the cerebellum, and between the caudate and the FOC among patients with AD compared with patients with MCI. We loosened the threshold for further study because the changing FCD value and FC intensity trends in the MCI-NC comparison were quite the opposite. Patients with MCI had increased FC intensity in the internal connections of the basal ganglia and the external connections between the basal ganglia and the cerebellum subregions. However, we found that the FC intensity of the internal and external connections of the basal ganglia—including the connection between the basal ganglia and the brain stem, between the putamen and the cerebellum, and between the basal ganglia and the lPFC, between the accumbens and the ACC, and between the accumbens and the hippocampus—among patients with MCI was lower in comparison with normal controls (voxel-level *p* < 0.01 and cluster-level *p* < 0.05, no correction) (Appendix A).

In graph theory analyses, no result was found in the comparisons of groups with correction, but the overall efficiencies of the bilateral caudate nucleus and the left amygdala among patients with AD were lower compared with patients with MCI, with the uncorrected threshold of *p* < 0.01. At the same threshold, a decreased degree was found in the bilateral caudate among patients with AD compared with patients with MCI (Figure 3b). In addition, we extracted the values of the overall efficiency and the degree values of the basal ganglia from two-sample *t* tests between patients with AD and those with MCI. As a result, we discovered some noteworthy differences in most subregions of the basal ganglia (Figure 3c).

### 3.4. Correlation among FCD, Graph Theory Indices, and Clinical Cognition Function Assessments

We discovered that both the FCD and the graph theory indices among patients with AD and those with MCI showed that the directional connection intensity of brain regions was significantly correlated with clinical cognitive function assessments (including MMSE and FAQ). In areas belonging to the basal ganglia (including the bilateral caudate and the right amygdala), there were positive correlations between the FCD (both short-range FCD and long-range FCD) and MMSE scores. In addition, we also found that the overall efficiency of the bilateral caudate nucleus, the left amygdala, and the right putamen had positive correlations with MMSE. Additionally, there was a positive correlation between the degree of the bilateral caudate, the left amygdala, and the right putamen and MMSE (Figure 4a).

FAQ scores were negatively correlated with short-range FCDs in the left caudate and the right amygdala. The correlation between the FAQ scores and long-range FCDs was also negative in the bilateral caudate, the right amygdala, and the right pallidum. Additionally, the correlations between the FAQ scores of the left caudate nucleus and the global efficiency were negative (Figure 4b).

However, there was no correlation between the FCD and NPI-Q scores, which was also the case for the graph theory indices. Since the GDS scores did not show significance and the CDR scores were not linear, the GDS score and the CDR score were not included in the correlation. In other brain regions, the FCD values and the local efficiencies showed no correlation with the cognition function assessments.

### 3.5. Classification Performances of FCD and Graph Theory Indices

To investigate the effect of the early detection of MCI, we analyzed the sensitivity and specificity of FCD and graph theory indices in specific brain regions among the MCI group and the normal controls. We learned that the maximum area under the curve (AUC) of the ROC in short-term FCDs was 0.793 in the right amygdala, and the sensitivity and the specificity were 81.3% and 75.0%, respectively. The largest AUC for the curve of long-range FCDs was 0.867, which was also located in the right amygdala, with a sensitivity of 87.5% and specificity of 75.0%. In addition, the AUC of the ROC of the overall efficiency of the right amygdala was 0.742 and was the largest, with a sensitivity of 68.8% and specificity of 87.5%. The AUC of the ROC was 0.703, shown in degrees, with a sensitivity of 50.0% and specificity of 87.5%. The ROC results showed no significant differences in local efficiency (Figure 5a).

To determine the influence of predicting cognitive disorder progression (from MCI to AD), we analyzed the sensitivity and specificity of the FCD and graph theory indices in specific regions for the AD and MCI groups. As a result, we discovered that, on the one hand, the AUC of the ROC in the short-range FCD was 0.797 in the left caudate, with a sensitivity of 75.0% and specificity of 75.0%; on the other hand, the most noteworthy region in the long-range FCD was the right amygdala, with an AUC of 0.879, a sensitivity of 87.5%, and specificity of 87.5%. Additionally, the maximum AUC of total efficiency ROC was 0.828, and was located in the left caudate, with a sensitivity of 87.5% and specificity of 62.5%. The local efficiency suggested that the AUC of the ROC was 0.736, with a sensitivity of 68.8% and a specificity of 75.0%. In addition, the largest AUC of the ROC was 0.805, shown in the left caudate, with a sensitivity of 75.0% and a specificity of 75.0% (Figure 5b).

In order to clarify the effect of AD diagnosis, we made full use of the statistics to analyze the sensitivity and specificity of the FCD and graph theory indices of specific regions in the AD group and normal controls; among these, the results suggest that the AUC for the ROC in the long-range FCD was 0.770 in the left caudate, the sensitivity was 93.8%, and the specificity was 62.5%. With a sensitivity of 62.5% and specificity of 93.7%, the AUC for the ROC of global efficiency in the right pallidum was the largest, at 0.781. Moreover, the AUC for the ROC of local efficiency in the right caudate was 0.730, with a sensitivity of 75.0% and specificity of 62.5%. The most prominent area was the right pallidum in degree, with an AUC of 0.762, and the sensitivity and specificity were 93.8% and 50.0%, respectively. Finally, there were no significantly different ROC results in the short-range FCD (Figure 5c).

All detailed results regarding the sensitivity and specificity of the ROC analyses are shown in Table 3.

## 4. Discussion

This study was first conducted to address the changes in exclusive functional connectivity in the basal ganglia during cognitive decline; accordingly, we utilized neuroimaging analyses and clinical statistics. In the current study, in addition to DMN-related regions, the basal ganglia, which involves the caudate, the amygdala, the putamen, and the globus pallidus, also showed changes in FCD values. In the connections between the basal ganglia nodes, we found aberrant functional connectivity strength and graph theory indices in both the internal and the external connections. There was a close correlation between FCDs and clinical impairment (assessed using MMSE scores and FAQ scores) in most of the mentioned regions and a similar correlation also existed between the graph theory indices and the clinical assessments. Furthermore, we discovered that the basal ganglia FCD values and the graph theory indices could predict the early detection of MCI and the progression of cognitive decline, and enable the diagnosis of AD. These findings may improve our understanding of AD and MCI pathology and provide a reference for future AD treatments.

Traditionally, changes in functional connectivity have been demonstrated in the basal ganglia among patients with Parkinson’s disease [12], epilepsy [13], and depression [14], but detailed analyses are lacking for AD. Many central basal ganglia areas are involved in cognitive functions, such as procedural learning and working memory [10,11,24]. Based on the cognitive and emotional functions which the basal ganglia is responsible for, similar trends in structural atrophy [6] and functional connectivity [7] of the basal ganglia subregions have been mentioned in previous studies. Generally speaking, most basal ganglia subregions exhibited significant differences in among-group comparisons of FCD and functional connectivity (FC) analyses. Notably, at the same statistical threshold, the increase in long-range FCDs in the amygdala was significant, whereas short-range FCDs were absent among patients with MCI, suggesting that neural compensation of the external connection in the amygdala and the prefrontal cortex may lead to sustained neuronal loss in the frontoparietal regions [25]. Conversely, with the same statistical threshold, we found a significant reduction in short-range FCDs in the caudate; however, we did not obtain results concerning long-range FCDs among patients with AD, which may indicate that the internal connection function of the basal ganglia is abnormal among patients with AD. Additionally, in further graph theory analyses, all results were obtained without correction, which may have something to do with the limited number of included subjects or the insufficient significance found in the basal ganglia functional connections of cognitive decline.

In this study, we delved into the functional connectivity of basal ganglia, focusing on the amygdala and striatum, the latter consisting of the dorsal medial striatum, dorsal lateral striatum, and ventral striatum [8,9]. The caudate and putamen performed essential roles in functional connectivity according to our results. However, under the present neuroimaging masks, the dorsal or ventral parts of the caudate and putamen are poorly demarcated, from which we would propose a reasonable hypotheses involving both dorsal and ventral parts. Dorsal medial striatum includes the dorsal part of the caudate, which was connected to the caudate–cerebellum dmPFC in previous studies. In response to this cognitive function of working memory, executing, and verbal fluency [26], some researchers have proposed that the most important part of memory decline in aging and cognitive impairment is working memory [27] and the possible mechanisms of striatal selective gating signals to PFC [28,29] is closely related to the dopaminergic system engaging into most cognitive processes [30]. In the following study, Zheng et al. [31] also confirmed the effect of the disruption of dorsal caudate connectivity in cognitive decline, which is consistent with the reduced FCD values and FC strengths between the caudate and the cerebellum in patients with AD of our study. However, increased FC strengths between the caudate and the cerebellum were observed in patients with MCI, similar to accumbens, the pallidum, and the amygdala. At the same time, also consistent with our results, the optimal compensation for neuron loss, proposed by Barrett et al. [32], may explain the theory of resilience of neural damage in the early stage of AD, the essential principle of which is to maintain the excitatory–inhibitory balance [33] in injured neurons. Thus, if inhibitory neurons are knocked down, the remaining inhibitory neurons will keep their signal representation error to increase their discharge rates independent of other neurons. Apart from that, owing to the similar cytoskeletal structure, structural connection, and development, the caudate and the putamen were regarded as new striata. In patients with MCI in our study, we found increased FCD values both in the caudate and putamen, while the reduction appeared in FC strengths between the caudate and the putamen, which may point out the coexistence of external neuron compensation and the internal functional recession of neostriatum in early cognitive decline.

We put emphasis on dorsal lateral striatum (DLS) and found increased FCD values and FC intensities in the putamen of patients with MCI. However, it is worth mentioning that, in addition to neural compensation mentioned above, the functional connectivity of the dorsal putamen, namely dorsal lateral striatum, should also be taken into consideration. Recently, the dorsal striatum which mediates procedural memories based on stimulus–response (S-R) associations was found by several researchers [10]. Aberrant DLS-dependent S-R memory occurs mainly in obsessive compulsive disorders [34] and drug addictions [35], but Goodman et al. [36] proposed a hypothesis that hippocampal damage [37] could be the key that is critical for the enhancing effects of emotional arousal in charge of the amygdala [38] on dorsal striatal memory. From the theory above, it is reasonable to deduce that hippocampus dysfunction in the early stages of AD triggers emotional regulation in the amygdala, which further enhances DLS-dependent S-R memory. Finally, the circuit conducts inputs to the cerebral cortex via pallidum transduction. In subsequent studies [39], an association between the dorsal striatum, amygdala, and dmPFC was also disclosed due to a mechanism called ‘threat of shock’, which may lead to the greater use of S-R memory. The dmPFC is a verified region of cognitive function and significant in the pathology of cognitive decline, and thus enhancing S-R memory (increased FC strength between the amygdala and the putamen, as well as increased FC strength between the putamen and the pallidum) in patients with MCI of our study is understandable. Furthermore, N-methyl-D-aspartate (NMDA) receptor induced synaptic plasticity and dopamine release may be a combined mechanism of DLS-dependent S-R memory and cognitive decline [11].

In our study, patients with AD were found to have reduced FCD values and FC intensities in the caudate nucleus, the amygdala, and the globus pallidus. The pallidum is the downstream output region to which most of the striatum is connected, making it easy to understand the drop in connectivity density when the functional connectivity of the striatum is disrupted. In addition, decreased FC strengths were found in the connectivity between the caudate and the cerebral cortex (including OFC and LPFC), the connectivity between the caudate and the amygdala, the connectivity between the caudate and the cerebellum, the connectivity between the accumbens and the ACC, and the connectivity between the accumbens and the brain stem among patients with cognitive decline. Combining these results, limbic system dysfunction with a disrupted vmPFC circuit [9,40] (including medial OFC and ACC), amygdala-VS (including accumbens, medial caudate), or posterior cerebellum/brain stem/pallidum may be used for explanation. In recent years, there has been a growing number of researchers paying attention to the effect of the limbic system (other than the hippocampus) on AD because of the common neuropsychiatric symptoms, such as emotional and social interaction which are dysregulated by AD [41]. Structural atrophy [6] and altered functional connectivity [41] in the limbic system have been demonstrated in previous studies. It has to be mentioned that the pathology of AD—similarly to Aβ deposition and tau-induced neurotoxicity—has been found in the limbic system as well [2,3]. Consistently, NMDA receptors are widely distributed around basal ganglia-related circuits and are essential in maintaining the functional stability of connections. Synaptic plasticity in the limbic systems of patients with AD may be disrupted due to the mechanism by which Aβ binds directly to NMDA receptor subunits [42] and competitively binds to modulate receptors (such as EphB2 receptors) to inhibit NMDA receptor binding [43].

Even though the impact of the basal ganglia on various neurological diseases has been extensively reported, we still accounted for it in our analyses of correlation with clinical assessments; additionally, we incorporated it into our ROC analyses of classification with the aim of confirming the impact of its functional connectivity changes on cognitive decline, as reflected in clinical manifestation. There is a close relationship between cognitive impairment (MMSE and FAQ) and FCD values in the basal ganglia of patients with AD, suggesting that declined functional connectivity of the basal ganglia is consistent with clinical manifestation. The results of the correlation between graph theory indices and clinical assessment add to the evidence that cognitive decline in AD is mainly associated with altered global functional connectivity rather than local functional connectivity. At the same time, the crucial role that the caudate holds in AD was also proven. In previous studies, ROC analysis was used to assess the predictive effects of neuroimaging biomarkers [44,45], but we merely took advantage of it to further verify the effects of the basal ganglia on clinical cognitive impairment in our study, because of its extensive effects on various neural diseases. Based on the FCD values and graph-theoretic metrics in the MCI assay, we identified the most important subregion of the basal ganglia, the amygdala, which was found to be consistent with the FCD results. Moreover, when it comes to cognitive decline progression and AD diagnosis, the caudate nucleus had the most prominent effect on the basal ganglia in most FCD values and graph-theoretic ROC analyses, again confirming the role of the caudate nucleus in the AD process. These findings may narrow the scope of pathological researching in cognitive decline related to the basal ganglia and promote research of targeted AD treatments.

There were several limitations in our study. First of all, the number of subjects was limited, so more clinical patients should be included in the future. In addition, we did not carry out clinical assessments—such as the Hamilton Anxiety Scale (HAMA) and the Hamilton Depression Scale (HAMD)—beyond cognitive evaluation, in spite of the diverse dysfunction in the basal ganglia of patients with AD. Finally, we did not apply machine learning in our evaluation of the accuracy of the results. We aim to introduce support vector machines or behavioral experiments to confirm the effect of the basal ganglia in further studies.

## 5. Conclusions

To our knowledge, this is the first study to exclusively observe the altered functional connectivity of the basal ganglia—using neuroimaging analyses and clinical statistics—in patients experiencing cognitive decline. The findings of this study could help us to build a better understanding of AD and MCI pathologies, as they relate to the basal ganglia, and may lead to new targets for AD therapies in further studies.

## Figures and Tables

**Figure 1 brainsci-12-01555-f001:**
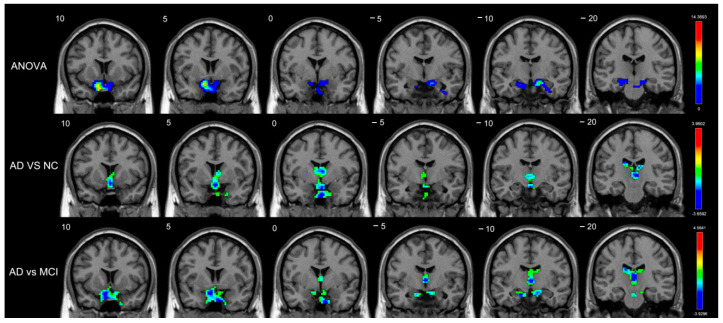
Brain regions with significant differences in short-range FCDs (voxel-level *p* < 0.01, cluster-level *p* < 0.05, GRF-corrected) among the AD, MCI, and NC groups. There were significant among-group differences in the short-range FCD in the right amygdala. Decreased short-range FCDs in patients with AD compared with normal controls were located in the bilateral caudate. Decreased short-range FCDs in patients with AD compared with patients with MCI were located in the bilateral caudate.

**Figure 2 brainsci-12-01555-f002:**
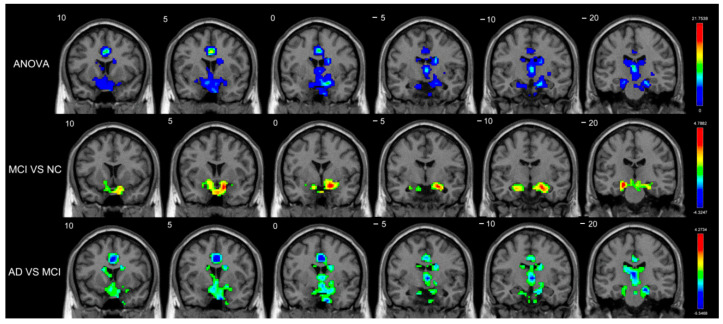
Brain regions with significant differences in long-range FCDs (voxel-level *p* < 0.01, cluster-level *p* < 0.05, GRF-corrected) among the AD, MCI, and NC groups. There were significant among-group differences of long-range FCDs in the bilateral caudate, the right amygdala, the right pallidum, and the right putamen. Increased long-range FCDs in patients with MCI compared with normal controls were located in the bilateral amygdala, the left putamen, and the right pallidum. Decreased long-range FCDs in patients with AD compared with patients with MCI were located in the bilateral caudate, the right amygdala, and the right pallidum.

**Figure 3 brainsci-12-01555-f003:**
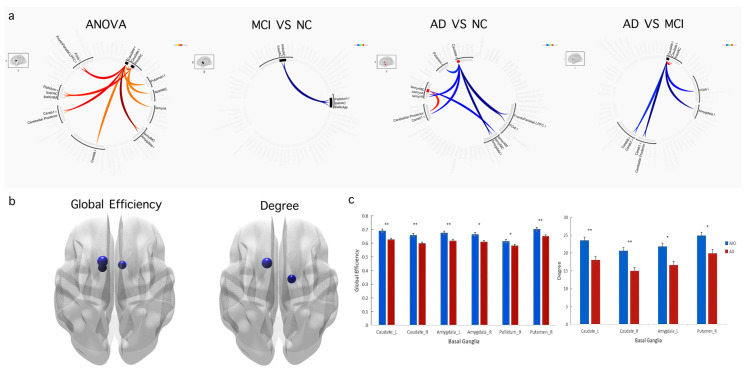
(**a**) Functional connectivity with significant differences (voxel-level *p* < 0.01, cluster-level *p* < 0.05, FDR-corrected) among the AD, MCI, and NC groups. (**b**) Graph theory analyses with significant differences (*p* < 0.01, no correction) among patients with AD and those with MCI. (**c**) Differences in graph theory indices of the basal ganglia between the patients with AD and those with MCI. (**—represents significant differences (*p* < 0.01); *—represents slight differences (*p* < 0.05)).

**Figure 4 brainsci-12-01555-f004:**
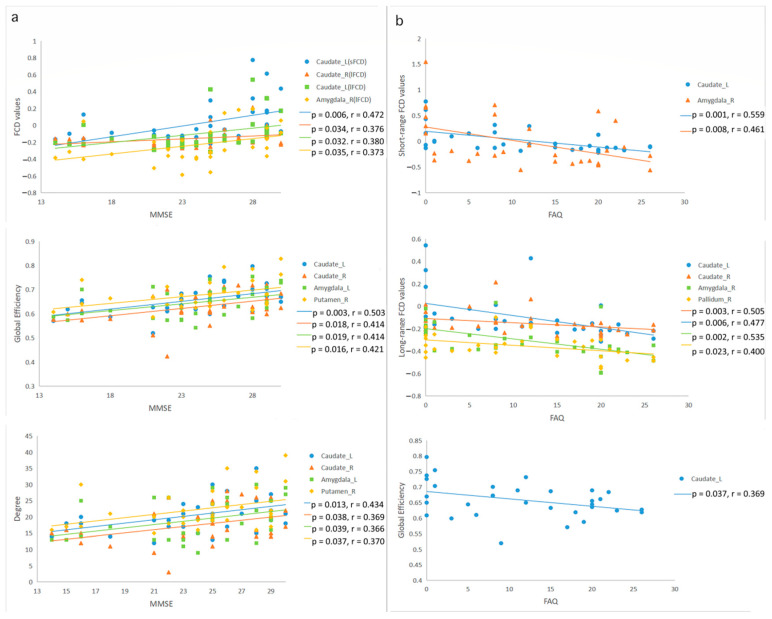
(**a**) Scatterplot of FCD and graph theory analyses in specific regions plotted against MMSE score. (**b**) Scatterplot of FCD and graph theory analyses in specific regions plotted against FAQ score.

**Figure 5 brainsci-12-01555-f005:**
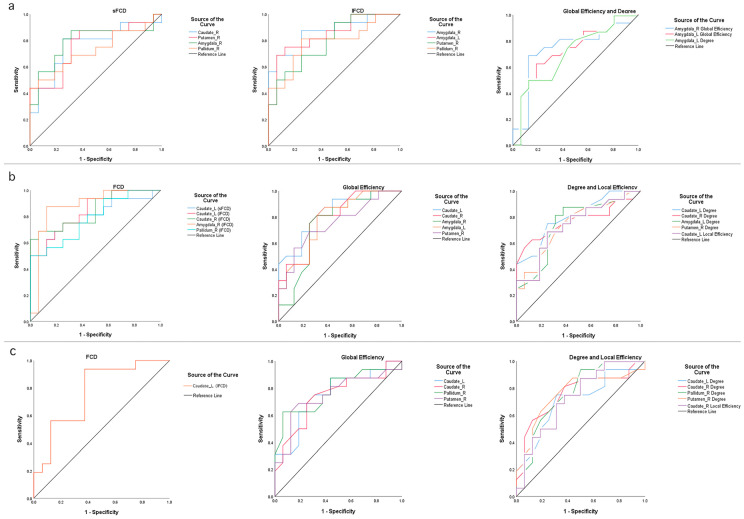
(**a**) Receiver operating characteristic (ROC) curve for FCDs and graph theory indexes in a between-group analysis of patients with MCI and normal controls. (**b**) Receiver operating characteristic (ROC) curve for FCDs and graph theory indices in a between-group analysis of patients with AD and those with MCI. (**c**) Receiver operating characteristic (ROC) curve for FCDs and graph theory indices in a between-group analysis of patients with AD and normal controls.

**Table 1 brainsci-12-01555-t001:** Demographic and clinical characteristics of the participants.

Characteristics	AD (*n* = 16)	MCI (*n* = 16)	NC (*n* = 16)	*p* Value
Age (years)	76.31 ± 3.98	75.56 ± 6.02	74.69 ± 4.66	0.653
Sex (female/male)	7/9	6/10	10/6	0.338
MMSE	20.75 ± 3.71	27.50 ± 1.79	29.19 ± 1.05	<0.001
GDS	1.31 ± 0.95	2.44 ± 3.44	0.56 ± 1.03	0.055
CDR	0.97 ± 0.34	0.50 ± 0.26	0.03 ± 0.13	<0.001
FAQ	17.81 ± 6.6	5.06 ± 5.43	0.06 ± 0.25	<0.001
NPI-Q	3.75 ± 3.21	3.31 ± 4.22	0.44 ± 0.73	0.008

*χ^2^*-test was used for sex; ANOVA was used for age and for MMSE, GDS, CDR, FAQ, and NPI-Q scores. AD—Alzheimer’s disease; MCI—mild cognitive impairment; NC—normal control; MMSE—Mini-Mental State Examination; GDS—Geriatric Depression Scale; CDR—Clinical Dementia Rating Scale; FAQ—Functional Activities Questionnaire; NPI-Q—Neuropsychiatric Inventory Questionnaire.

**Table 2 brainsci-12-01555-t002:** FCDs among-group comparison in the AD, MCI, and NC groups (voxel-level *p* < 0.01 and cluster-level *p* < 0.05, GRF-corrected).

Indices	Region	Brodmann’s Area	MNI Coordinates	*T* Value	Cluster(Voxels)
ANOVA					
Short-range	Amygdala_R	BA34	(18,0,−17)	5.9018	3
Long-range	Caudate_R	-	(18,−5,20)	11.3438	66
	Caudate_L	-	(−16,−12,23)	6.4713	43
	Amygdala_R	BA34	(19,1,−17)	13.2933	32
	Pallidum_R	BA48	(19,0,−5)	7.8705	14
	Putamen_R	BA48	(33,−9,0)	6.1615	7
MCI vs. NC					
Long-range	Amygdala_R	BA34	(21,−2,−17)	4.4157	38
	Amygdala_L	-	(−13,2,−15)	3.3582	6
	Putamen_L	BA25	(−14,7,−9)	3.2804	7
	Pallidum_R	BA25	(12,3,−5)	3.1270	5
AD vs. NC					
Short-range	Caudate_L	-	(−7,1,11)	−2.9712	29
	Caudate_R	-	(9,5,7)	−2.8622	8
AD vs. MCI					
Short-range	Caudate_L	-	(−18,−21,23)	−3.2570	9
	Caudate_R	-	(16,−18,21)	−2.8545	4
Long-range	Caudate_L	-	(−15,14,18)	−3.7649	66
	Caudate_R	-	(17,6,23)	−3.7601	54
	Amygdala_R	BA48	(19,2,−11)	−3.6551	20
	Pallidum_R	BA48	(19,−3,−2)	−3.4747	9

**Table 3 brainsci-12-01555-t003:** Classification performances of FCD and graph theory values (*p* < 0.05).

Conditions	Regions	*p* Value	AUC Value	Sensitivity	Specificity
NC–MCI					
Short-range FCD	Caudate_R	0.012	0.762	81.3%	75.0%
	Amygdala_R	0.005	0.793	81.3%	75.0%
	Putamen_R	0.010	0.766	87.5%	62.5%
	Pallidum_R	0.029	0.727	43.8%	100.0%
Long-range FCD	Amygdala_R	0.000	0.867	87.5%	75.0%
	Amygdala_L	0.001	0.844	75.0%	87.5%
	Putamen_L	0.005	0.793	68.8%	75.0%
	Pallidum_L	0.006	0.785	81.3%	75.0%
Global Efficiency	Amygdala_R	0.019	0.742	68.8%	87.5%
	Amygdala_L	0.027	0.729	62.5%	81.2%
Degree	Amygdala_L	0.050	0.703	50.0%	87.5%
MCI–AD					
Short-range FCD	Caudate_L	0.004	0.797	75.0%	75.0%
Long-range FCD	Caudate_L	0.001	0.840	75.0%	75.0%
	Caudate_R	0.001	0.848	68.8%	93.7%
	Amygdala_R	0.000	0.879	87.5%	87.5%
	Pallidum_R	0.006	0.785	50.0%	100.0%
Global Efficiency	Caudate_L	0.002	0.828	87.5%	62.5%
	Caudate_R	0.004	0.795	81.3%	68.7%
	Amygdala_R	0.018	0.746	81.3%	68.7%
	Amygdala_L	0.006	0.783	87.5%	62.5%
	Putamen_R	0.016	0.750	68.8%	75.0%
Local Efficiency	Caudate_L	0.023	0.736	68.8%	75.0%
Degree	Caudate_L	0.003	0.805	75.0%	75.0%
	Caudate_R	0.009	0.771	62.5%	87.5%
	Amygdala_L	0.017	0.748	81.3%	68.7%
	Putamen_R	0.020	0.740	81.3%	56.2%
NC–AD					
Long-range FCD	Caudate_L	0.009	0.770	93.8%	62.5%
Global Efficiency	Caudate_L	0.014	0.754	68.8%	75.0%
	Caudate_R	0.020	0.740	68.8%	75.0%
	Pallidum_R	0.007	0.781	62.5%	93.7%
	Putamen_R	0.017	0.748	68.8%	81.2%
Local Efficiency	Caudate_R	0.026	0.730	75.0%	62.5%
Degree	Caudate_L	0.044	0.709	75.0%	68.7%
	Caudate_R	0.013	0.758	81.3%	62.5%
	Pallidum_R	0.012	0.762	93.8%	50.0%
	Putamen_R	0.013	0.758	68.8%	75.0%

## Data Availability

The data that support the findings of this investigation are available in the ADNI database (http://adni.loni.usc.edu (accessed on 2 February 2022)). As such, the investigators within the ADNI contributed to the design and implementation of ADNI and/or provided data but did not participate in analysis or writing of this report. A complete listing of ADNI investigators can be found at: http://adni.loni.usc.edu/wp-content/uploads/how_to_apply/ADNI_Acknowledgement_List.pdf (accessed on 2 February 2022). The authors’ data are available upon reasonable request and with ADNI’s approval.

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
