# Peer review of "Altered Functional Connectivity of Basal Ganglia in Mild Cognitive Impairment and Alzheimer’s Disease"

_brainsci, 2022, doi:10.3390/brainsci12111555_

Round 1
Reviewer 1 Report
The authors examine the functional connectivity of the basal ganglia in people with MCI and AD. Overall, this manuscript is noteworthy as it contributes significantly to the understanding of the role of altered functional connectivity of the basal ganglia in cognitive decline and adds to the existing literature that focuses mainly on the role of changes in functional connectivity of the DMN network. I believe that the topic of this study may be of great interest to readers of Brain Science. I believe the following minor revisions need to be made:
- the abstract refers to ADNI patients (line 21) without specifying what this acronym means
- line 194: the correct term is sex and not gender
- throughout the manuscript when p=.000 it would be more correct to write p <.001. Both technically and conceptually, p-values cannot equal 0.
- The authors correctly report that many central basal ganglia areas are involved in cognitive functions such as procedural learning and working memory and cite the relevant literature. Since there is also literature that has hypothesized and highlighted the involvement of the basal ganglia in complex circuits responsible for recursion in action and language, I think it is important that the authors also refer to this literature (Vicari & Adenzato PMID: 24762973), since the results of their study are consistent with the hypothesis of a role for the basal ganglia not only in procedural learning and working memory, but also in recursion. This is an added value of the present work.
- The manuscript needs to be revised in English by a native speaker
Reviewer 2 Report
Here, Xiong et al. demonstrated that functional connectivity of basal ganglia underwent changes in Alzheimer's Disease and Mild Cognitive Impairment patients by analyzing fMRI data through functional connectivity and graph theory methods. However, there are major issues of the manuscript including:
1. Representation of majority of major figures, particularly including Figure 3, 4 and 5, are with extremely low quality. Panels within these figures are unreadable and incomprehensive. There is no way to read the captions and to evaluate the information and conclusion that the author made through the relevant paragraphs. Figures with much higher resolution and with clear captions need to be included and shown in the manuscript for future evaluation.
2. P value by statistical analysis was not appropriately demonstrated in the Table 1. Extremely small P value should be shown as "P<0.001" or "P<0.0001" rather than "P=0.000".
Also, several minor issues need to be changed as well:
1. in the abstract, abbreviations including DMN and FCD need to be explained at the first sites of occurrence.
Round 2
Reviewer 2 Report
The authors have addressed all the questions that reviewers put forward during the first round revision. Approve to be accepted and published.